# Winter is coming: Pathogen emergence in seasonal environments

**Philippe Carmona** [1], **Sylvain Gandon** [2]*

**1** Laboratoire de Mathématiques Jean Leray, Université de Nantes, Nantes, France, **2** CEFE, CNRS, Univ Montpellier, Univ Paul Valéry Montpellier 3, EPHE, IRD, 34293 Montpellier Cedex 5, France

☉ These authors contributed equally to this work.
* sylvain.gandon@cefe.cnrs.fr

**Data Availability Statement:** All relevant data are within the manuscript and its Supporting Information files.

**Funding:** Philippe Carmona acknowledges funding from C.N.R.S which supported a 50% delegation in 2019 to the CEFE in Montpellier, as a visiting

## Abstract

Many infectious diseases exhibit seasonal dynamics driven by periodic fluctuations of the environment. Predicting the risk of pathogen emergence at different points in time is key for the development of effective public health strategies. Here we study the impact of seasonality on the probability of emergence of directly transmitted pathogens under different epidemiological scenarios. We show that when the period of the fluctuation is large relative to the duration of the infection, the probability of emergence varies dramatically with the time at which the pathogen is introduced in the host population. In particular, we identify a new effect of seasonality (the *winter is coming* effect) where the probability of emergence is vanishingly small even though pathogen transmission is high. We use this theoretical framework to compare the impact of different preventive control strategies on the average probability of emergence. We show that, when pathogen eradication is not attainable, the optimal strategy is to act intensively in a narrow time interval. Interestingly, the optimal control strategy is not always the strategy minimizing $R_0$, the basic reproduction ratio of the pathogen. This theoretical framework is extended to study the probability of emergence of vector borne diseases in seasonal environments and we show how it can be used to improve risk maps of Zika virus emergence.

## Author summary

Seasonality drives fluctuations in the probability of pathogen emergence, with dramatic consequences for public health and agriculture. We show that this probability of pathogen emergence can be vanishingly small *before* the low transmission season. We derive the conditions for the existence of this *winter is coming* effect and identify optimal control strategies that minimize the risk of pathogen emergence. We generalize this framework to account for different forms of environmental variations, different modes of control and complex pathogen life cycles. We illustrate how this framework can be used to improve predictions of Zika emergence at different points in space and time.

professor. He also thanks the Centre Henri Lebesgue ANR-11-LABX-0020-01 for creating an attractive mathematical environment. This project has received financial support from the CNRS through the 80|Prime program. The funders had no role in study design, data collection and analysis, decision to publish, or preparation of the manuscript.

## Introduction

The development of effective control strategies against the emergence or re-emergence of pathogens requires a better understanding of the early steps leading to an outbreak [1, 2, 3, 4]. Classical models in mathematical epidemiology predict that whether or not an epidemic emerges depends on $R_0 = \frac{\lambda}{\mu}$ the basic reproduction ratio of the pathogen, where $\lambda$ is the *birth rate* of the infection (a function of the transmission rate and the density of susceptible hosts) and $\mu$ is the *death rate* of the infection (a function of the recovery and mortality rates). In the classical deterministic description of disease transmission, the pathogen will spread if $R_0 > 1$ and will go extinct otherwise (Fig 1). This deterministic description of pathogen invasion relies on the underlying assumption that the initial number of introduced pathogens is large. The early stages of an invasion are, however, typically characterized by a small number, $n$, of infected hosts. These populations of pathogens are thus very sensitive to demographic stochasticity and may be driven to extinction even when $R_0 > 1$. The probability of emergence $p_e^n$

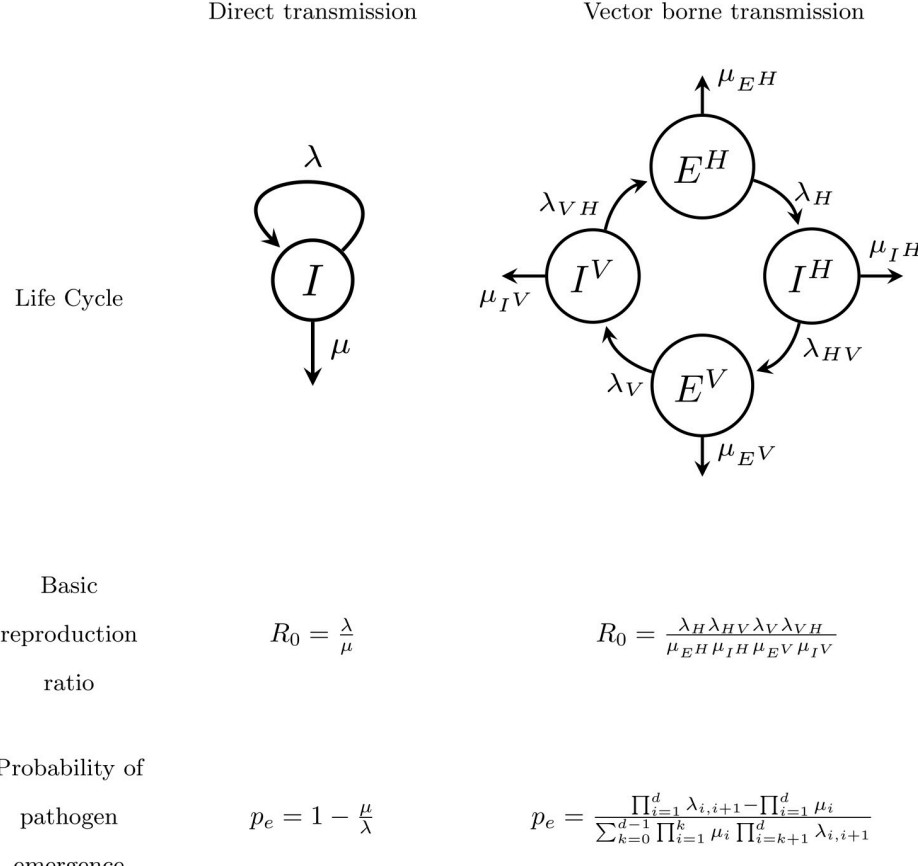

| | Direct transmission | Vector borne transmission |
|---|---|---|
| Life Cycle | | |
| Basic reproduction ratio | $R_0 = \frac{\lambda}{\mu}$ | $R_0 = \frac{\lambda_H \lambda_{HV} \lambda_V \lambda_{VH}}{\mu_{EH} \mu_{IH} \mu_{EV} \mu_{IV}}$ |
| Probability of pathogen emergence | $p_e = 1 - \frac{\mu}{\lambda}$ | $p_e = \frac{\prod_{i=1}^{d} \lambda_{i,i+1} - \prod_{i=1}^{d} \mu_i}{\sum_{k=0}^{d-1} \prod_{i=1}^{k} \mu_i \prod_{i=k+1}^{d} \lambda_{i,i+1}}$ |

**Fig 1. Transmission mode and pathogen emergence without seasonality.** In a direct transmission model pathogen dynamics is driven by the birth rate $\lambda$ and the death rate $\mu$ of a single infected compartment $I$. In a vector borne transmission model pathogen dynamics is driven by the birth rates and death rates of multiple compartments: exposed and infected humans ($E^H$, $I^H$), exposed and infected mosquito vectors ($E^V$, $I^V$). In the absence of seasonality (i.e. no temporal variation in birth and death rates) the basic reproduction ratio $R_0$ can be expressed as a ratio between birth and death rates. The probability of emergence $p_e$ after the introduction of a single infected individual can also be expressed as a function of these birth and death rates. With vector borne transmission this probability of emergence depends on which infected host is introduced (Figure E in S1 Text). Here we give the probability of emergence after the introduction of a single human exposed to the pathogen, $E^V$, and where the index $i$ refers to the four consecutive states of the pathogen life cycle (see sections 2 and 3 of S1 Text).

refers to the probability that, after the introduction of $n$ infected hosts, a non-evolving pathogen avoids initial extinction and leads to an epidemic. The analysis of stochastic epidemic models and the derivation of the probability of a major epidemic can be traced back to the work of Bailey (1953) and Whittle (1955). Under the reasonable assumption that the initial spread of directly transmitted disease follows a one dimensional birth-death branching process the probability of emergence is zero when $R_0 < 1$ and, when $R_0 > 1$, it is equal to [3, 4, 5, 6, 7]:

$$p_e^n = 1 - \left(\frac{1}{R_0}\right)^n.$$

(1)

The above results rely on the assumption that birth and death rates of the infection remain constant through time (i.e. time homogeneous branching process). Many pathogens, however, are very sensitive to fluctuations of the environment. For instance, the fluctuations of the temperature and humidity have been shown to have a huge impact on the infectivity of many viral pathogens like influenza [8] and a diversity of other infectious diseases [9, 10]. In addition, many pathogens rely on the presence of arthropod vectors for transmission and the density of vectors is also very sensitive to environmental factors like temperature and humidity [11]. To account for these environmental variations, the birth and death rates are assumed to be functions of time: $\lambda(t)$ and $\mu(t)$, respectively. The basic reproduction number is harder to compute but the probability of emergence $p_e(t_0)$ when one infected individual is introduced (i.e. $n = 1$) at time $t_0$ is well known (see e.g. [12] or [13, Chapter 7]):

$$p_e(t_0) = \frac{1}{1 + \int_{t_0}^{+\infty} \mu(t) e^{-(\varphi(t) - \varphi(t_0))} \, dt},$$

(2)

with $\varphi(t) := \int_0^t r(s) \, ds$ where $r(t) = \lambda(t) - \mu(t)$ is the Malthusian growth rate of the pathogen population at time $t$ (another derivation of (2) is given in section 2.5 of S1 Text). Because we are interested in seasonal variation we can focus on periodic scenarios where both $\lambda_T(t)$ and $\mu_T(t)$ have the same period $T$, one year. In this case, the basic reproduction number has been computed in [3, 14] as the spectral radius of the next generation operator, and is the ratio of time averages of birth and death rates:

$$R_0 = \frac{\bar{\lambda}}{\bar{\mu}}, \quad \text{with} \quad \bar{\lambda} = \frac{1}{T} \int_0^T \lambda_T(s) \, ds, \bar{\mu} = \frac{1}{T} \int_0^T \mu_T(s) \, ds.$$

(3)

When $R_0 < 1$ the pathogen will never produce major epidemics and will always be driven to extinction. When $R_0 > 1$, however, a pathogen introduced at a time $t_0$ may escape extinction. In this case the probability of emergence can also be expressed as a ratio of average birth and death rates, but with different weights (see section *Pathogen emergence with seasonality* of Methods):

$$p_e(t_0, T) = 1 - \frac{\int_0^T \mu_T(s + t_0) e^{-\varphi_T(s + t_0)} \, ds}{\int_0^T \lambda_T(s + t_0) e^{-\varphi_T(s + t_0)} \, ds}.$$

(4)

Note that this quantity refers to the probability of major epidemics, the probability that the pathogen population does not go extinct. Minor epidemics are likely to outburst if the pathogen is introduced during the high transmission season but those outbreaks do not count as major epidemics if they go extinct during the low transmission seasons.

In the following we show that very good approximations of the probability of pathogen emergence can be derived from this general expression when the period is very large (or very small) compared to the duration of the infection. These approximations give important insights on the effect of the speed and the shape of the temporal fluctuations of the environment on the probability of pathogen emergence. We use this theoretical framework to determine optimal control strategies that minimize the risk of pathogen emergence. We provide clear cut recommendations in a range of epidemiological scenarios. We also show how this theoretical framework can be extended to account for the effect of seasonality in vector borne diseases. More specifically, we use this model to estimate the probability of Zika virus emergence throughout the year at different geographic locations.

## Results

### Emergence of directly transmitted pathogens

For the sake of simplicity we start our analysis with a directly transmitted disease with a constant clearance rate $\mu(t) = \mu$, but with seasonal fluctuations of the transmission rate, $\lambda(t)$. This epidemiological scenario may capture the seasonality of many infectious diseases. For instance, increased contact rates among children during school terms has been shown to have a significant impact on the transmission of many childhood infections [15, 16]. Seasonal fluctuations in temperature and humidity can also drive variations in the survival rate of many viruses and result in seasonal variations in transmission rates [17, 18].

Both the speed and the amplitude of the fluctuations of $\lambda(t)$ can affect the probability of pathogen emergence. Yet, when the period $T$ of the fluctuations is short compared to the duration $1/\mu$ (e.g. fluctuations driven by diurnal cycles are fast), the probability of pathogen emergence can be approximated by (Fig 2E and 2F, see section *Asymptotics for small periods* of Methods):

$$p_e \approx 1 - \bar{\mu}/\bar{\lambda} = 1 - 1/R_0 \tag{5}$$

In other words, the probability of emergence does not depend on the timing of the introduction event and it is only driven by the average transmission rate.

When the fluctuations are slower, however, the probability of pathogen emergence does depend on the timing of the introduction. The probability of emergence drops with the transmission rate (Fig 2E and 2F). When the period of the fluctuation is long, a natural approximation is (see section *Asymptotics for large periods* of Methods):

$$p_e(t_0) \approx 1 - \mu(t_0)/\lambda(t_0) \tag{6}$$

This is a very good approximation whenever the birth rate of the infection remains higher than the death rate throughout the year (i.e., $\lambda(t) > \mu(t)$, Fig 2). However, when $\lambda(t)$ can drop below $\mu(t)$, the above approximation fails to capture the dramatic reduction of the probability of emergence occurring at the end of the high transmission season. When the introduction time of the pathogen is shortly followed by a low transmission season, the introduced pathogen is doomed because it will suffer from the bad times ahead (see section *Asymptotics for large periods* of Methods). We call this the *winter is coming* effect. Fig 2D and 2F provide a geometric interpretation of this effect. In the low transmission season the integrated growth rate $\varphi(t)$ drops with $t$ because the Malthusian growth rate of the pathogen is negative. Any epidemic starting during (dark gray shading) or just before (light gray shading) this period is unlikely to escape extinction because of this demographic trap. We further explore this effect in Figure A in S1 Text, under different types of seasonal variations: square waves and sinusoidal waves. As

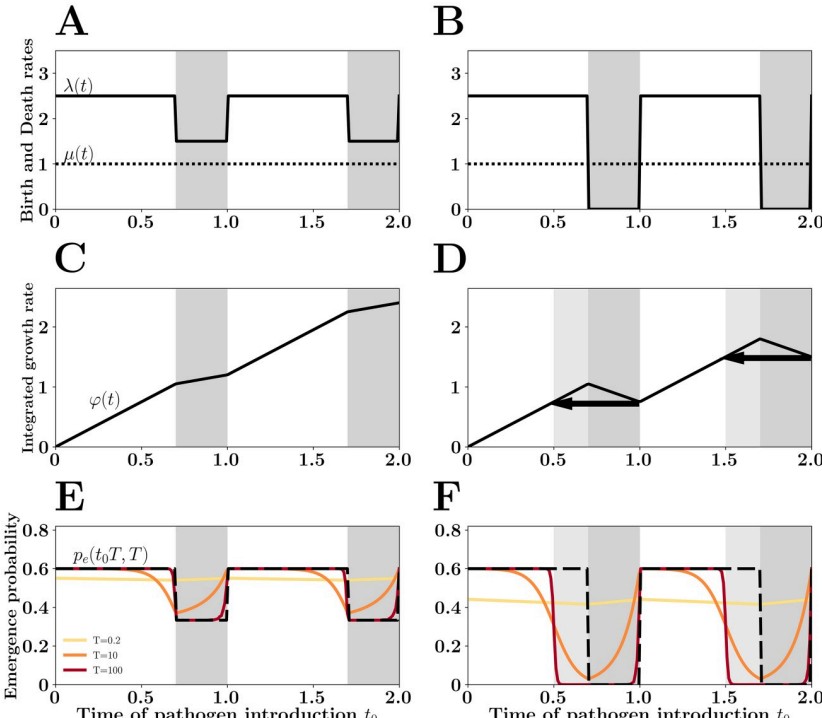

**Fig 2. The *winter is coming* effect.** Pathogen *birth rate* (i.e. transmission rate) $\lambda(t)$ is assumed to vary periodically following a square wave (A and B). During a portion $1 - \gamma$ of the year transmission is maximal ($\gamma = 0.7$ in this figure) and $\lambda(t) = \lambda_0$. In the final portion of the year $\lambda(t)$ drops (low transmission season in gray). In A $\lambda(t)$ varies between $\lambda_0 = 2.5$ and 1.5 and, in B $\lambda(t)$ varies between $\lambda_0 = 2.5$ and 0. Pathogen *death rate* $\mu(t)$ (a function of recovery and death rates of the infected host) is assumed to be constant and equal to 1 in this figure. When the net growth rate of the pathogen remains positive in the low transmission season ($\lambda(t) > \mu(t)$, A, C and E) the probability of emergence of a pathogen introduced at time $t_0$ can be well approximated by Eq (6): $p_e(t_0) \simeq 1 - \frac{\mu(t_0)}{\lambda(t_0)}$ (dashed line in E and F) if the duration of the infection is short relative to the period $T$ of the fluctuation (E). In contrast, if the low transmission season is more severe ($\lambda(t) < \mu(t)$, B, D and F), the negative growth rate $\varphi(t)$ of the pathogen population during this period creates a demographic trap and reduces the probability of emergence at the end of the high transmission season. This *winter is coming* effect is indicated with black arrow in (D) and with the light gray shading in (D) and (F). This effect is particularly pronounced when the period of the fluctuations of the environment is large relative to the duration of the infection (i.e., when $T$ is large, F). When the period $T$ of the fluctuation is small relative to the duration of the infection, the probability of emergence is well approximated by Eq (5): $p_e \simeq 1 - \frac{1}{R_0}$ whatever the time of pathogen introduction (in A, $R_0 = 2.2$ and $p_e \simeq 0.55$; in B, $R_0 = 1.75$ and $p_e \simeq 0.43$).

expected, the *winter is coming* effect is particularly pronounced when the period of the fluctuations are long relative to the duration of the infection (Fig 2F).

**Optimal control.** Our theoretical framework can be used to identify optimal control strategies. The objective is to minimize the average probability of emergence under the assumption that the introduction time is uniformly distributed over the year:

$$\langle p_e \rangle := \int_0^1 p_e(t_0) \, dt_0$$

Control is assumed to act via an instantaneous reduction $\rho(t)$ of the transmission rate of the pathogen: $\lambda_\rho(t) = \lambda(t)(1 - \rho(t))$. We also assume that higher control intensity is costly and we define the cost of a given control strategy as a function of the intensity and the duration of the

control:

$$C := \int_0^1 \rho(s)\,ds$$

More explicitly, we assume that the control strategy is governed by three parameters: $t_1$ and $t_2$, the times at which the control starts and ends, respectively, and $\rho_M$ the intensity of control during the interval $[t_1, t_2]$. The cost of such a control strategy is thus: $C = (t_2 - t_1)\rho_M$. For a given investment in disease control $C$, what are the values of $t_1$, $t_2$ and $\rho_M$ that minimize the average probability of emergence $\langle p_e \rangle$?

We first answer this question when the fluctuation of transmission is a square wave where $\lambda(t)$ oscillates between $\lambda_0$ (for a fraction $1 - \gamma$ of the year) and 0, while $\mu(t) = 1$ throughout the year (Fig 3). For instance, such periodicity may be driven by school terms with high transmission between students when school is on and low transmission when school is off [19]. The basic reproduction after control in the high transmission period is equal to $R_0 - C$ (see section 1 of S1 Text). In other words, under this scenario, when the investment in control reaches a threshold (i.e. when $C > R_0 - 1$) the basic reproduction (after control) of the pathogen drops below one and the probability of emergence vanishes. Fig 3 shows how $\langle p_e \rangle$ varies with different types of interventions when this level of control is unattainable (e.g. because the value of $R_0$ is too high). We assume that the investment in control is fixed and equal to $C = (t_2 - t_1)\rho_M = 0.2$ and we explore how different values of $t_1$ and $\rho_M$ affect $\langle p_e \rangle$. A naive strategy where control is

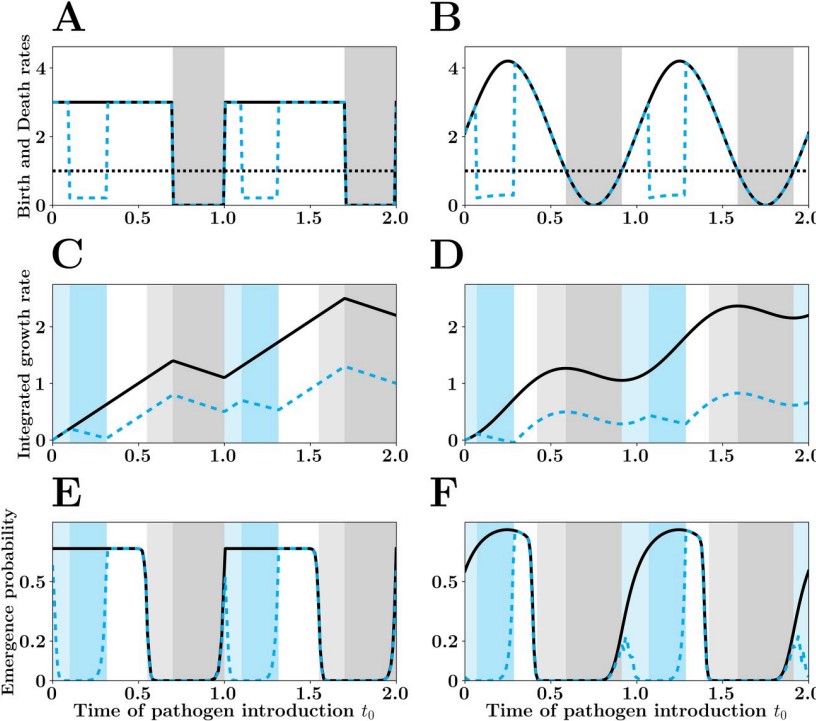

**Fig 3. Optimal Control for square wave (A, C and E) and sinusoidal birth rates (B, D and F).** In A and B we plot The pathogen birth rate before (black line) and after the optimal control (dashed blue line) which minimizes the mean emergence probability $< p_e >$ (see also Fig 4). The square wave assumes that $\lambda(t) = 3 \; \mathbf{1}_{(0<t<0.7)}$. The sinusoidal wave assumes that $\lambda(t) = 2(1 + \sin(2\pi t))$. As in Fig 2 the gray shadings refers to the low transmission season (gray) and the *winter is coming* effect (light gray). Similarly, we indicate the additional low transmission period induced by control (blue shading) and the additional *winter is coming* effect induced by control (light blue shading).

applied throughout the high transmission season ($t_1 = 0$, $t_2 = 0.7$, $\rho_M = 2/7$) yields an average probability of emergence equal to $\langle p_e \rangle = 0.233$. Many alternative strategies where the control is applied more intensely but in a limited portion of the high transmission season (Fig 3A, 3C and 3E and S2) yield lower values of $\langle p_e \rangle$. In particular, all the strategies that fall within the dotted red curve of Fig 4A have $\langle p_e \rangle = 0.166$. Indeed, all the strategies that fall in this region maximize the *winter is coming* effect. Fig 3C and 3E show how the timing of control (indicated by the blue shading) for one of these optimal strategies minimizes the probability of emergence via an extension of the effects of the low transmission season.

Second, we consider a seasonal environment where $\lambda(t)$ follows a sinusoidal wave, while $\mu(t) = 1$ throughout the year (Fig 3B, 3D and 3F). Such periodicity may arise with more gradual changes of the abiotic environment driven by climatic seasonality [19]. Under this scenario, pathogen transmission varies continuously and the basic reproduction after control does depend on the time at which control is applied. The basic reproduction ratio is minimized when the intensity of control is maximal ($\rho_M = 1$) in a time interval centered on the time at which pathogen transmission reaches its peak (red cross in Fig 4B). In contrast, the optimal control strategy that minimizes $\langle p_e \rangle$ starts earlier, lasts longer and is a bit less intense (blue cross in Fig 4B). As discussed in the square wave scenario, the timing of control in the optimal strategy extends the *winter is coming* effect. Fig 3D and 3F show that the optimal strategy (indicated by the blue shading) prolongs the effect of the low transmission season.

## Emergence of vector borne pathogens

Next we want to expand the above analysis to a more complex pathogen life cycle. Indeed, many emerging pathogens are vector borne [1, 20] and the probability of pathogen emergence can also be computed under this life cycle [6, 7, 21, 22]. Arboviruses, for instance, use different mosquito species as vectors and are responsible for major emerging epidemics in human populations [23]. In the following, we use a classical epidemiological model of Zika virus

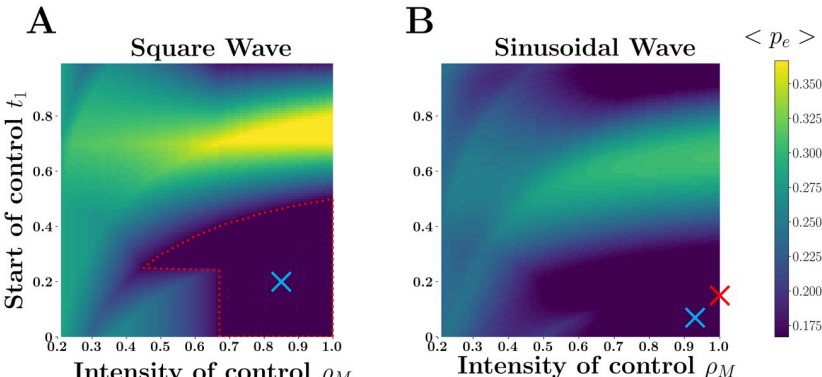

**Fig 4. Mean probability of pathogen emergence for different control strategies with (A) square wave and (B) sinusoidal wave fluctuations.** We used the same scenarios as in Fig 3 and we fix the investment in control (cost of control $C = \rho_M(t_2 - t_1) = 0.2$). We explore how the intensity of control ($\rho_M$) and the timing of control (between $t_1$ and $t_2$) affect $\langle p_e \rangle$, the mean probability of pathogen emergence (lighter shading refers to higher values of $\langle p_e \rangle$). For the square wave scenario we identify a range of optimal strategies withing the dotted red curve where $\langle p_e \rangle$ is minimized. The optimal strategies used in Fig 3 are indicated with a blue cross for both the square wave (A) and the sinusoidal wave (B). The minimal and maximal value for $\langle p_e \rangle$ are: $0.166 - 0.366$ (square wave) and $0.085 - 0.31$ (sinusoidal wave). For the square wave (A), $R_0 = 1.5$ does not depend on the timing and the intensity of the control. For the sinusoidal wave (B), there is a single strategy minimizing $R_0$, namely $R_0 = 1.28$ for $t_1 = 0.15$ and $\rho_M = 1.0$, marked with a red cross in B. With the sinusoidal wave there is a single control strategy minimizing $\langle p_e \rangle$ for $t_1 = 0.07$ and $\rho_M = 0.93$ (blue cross in B).

transmission which has been parameterized using empirical data sets to determine the probability of emergence under various regimes of seasonality (see section 3 of S1 Text). In this model, the pathogen may appear in four different states (Fig 1): exposed and infectious mosquitoes ($E^V$ and $I^V$), exposed and infectious humans ($E^H$ and $I^H$). The stochastic description of this epidemiological model yields a four dimension multi-type birth-death branching process (see section 2 of S1 Text). In the absence of seasonality (homogeneous case) the basic reproduction ratio of the pathogen is the ratio of the product of birth rates by the product of the death rates (Fig 1). The probability of emergence after the introduction of a single infected host in state $\in (E^H, I^H, E^V, I^V)$:

$$p_e = \frac{\prod_{i=i}^{d} \lambda_{i,i+1} - \prod_{i=1}^{d} \mu_j}{\sum_{k=0}^{d-1} \prod_{i=1}^{k} \mu_i \prod_{i=k+1}^{d} \lambda_{i,i+1}} \qquad (7)$$

where the index $i$ refers to the $d$ consecutive states of the pathogens, starting with the state in which the pathogen is introduced. Hence $\lambda_{i,i+1}$ denotes the *birth rate* of an infection in state $i + 1$ from an infection in state $i$, and $\mu_i$ denotes the *death rate* of an infection in state $i$. Note that the state of the introduced infection can have a huge impact on the probability of pathogen emergence (Figure E in S1 Text). One may expect that if the epidemic starts in a bad quality host with a low $R_i = \frac{\lambda_{i,i+1}}{\mu_i}$ ratio the pathogen is more likely to go extinct than if it starts with a good quality host (with a high $R_i$ ratio). We show in the S1 Text subsection 2.2 that this is indeed the case in dimension $d = 2$ (see also [21, 22]). But things become more complex when $d > 2$ because the quality of the following hosts in the transmission cycle matter as well. In other words, we can observe a *weak host is coming* effect on the probability of emergence. This effect is akin to the *winter is coming* effect that we discuss above, but it is driven by the alternation of the quality of hosts, not by seasonality.

Seasonality can drive pathogen transmission through the fluctuations of the available density of the mosquito vector. Following [24] we assume that mosquito density fluctuates with temperature and is maximal at $T_{opt}$, the optimal temperature for mosquito reproduction (see sup info). The rate $\lambda_{I^H, E^V}$ at which mosquitoes are exposed to the parasite is directly proportional to $N_V/N_H$. In such a fluctuating environment the $R_0$ is the spectral radius of the next generation operator, see [3, 14] but there is no analytic expression for $R_0$. Yet, it is tempting to use Eq (7) with the birth and death rates functions of the introduction time $t_0$, to obtain an approximation $p_e(t_0)$ for large periods. The exact probability of emergence can be efficiently computed numerically thanks to the seminal work of [25]. Fig 5 explores the difference between this naive expectation and the exact value of the probability of emergence. Crucially, we recover the same qualitative patterns observed in the direct transmission model. In particular, we notice that when the product of birth rates remains higher than the product of death rates the naive expectation for the probability of emergence is not too far from the exact value of $p_e(t_0)$. However, when seasonality induces more pronounced drops in transmission, we recover the *winter is coming* effect where the probability of emergence can be very low before the low transmission season (Fig 5D). It is also possible to identify numerically the optimal control strategies minimizing the probability of Zika emergence (Figure F in S1 Text).

## Discussion

The effect of seasonality on the probability of pathogen emergence depends critically on the duration of the infection $1/\mu$ relative to the period $T$ of the fluctuation. When the period of the fluctuation is small (i.e., $T < 1/\mu$) the environment changes very fast and the probability of emergence does not depend on the timing of pathogen introduction but on the average

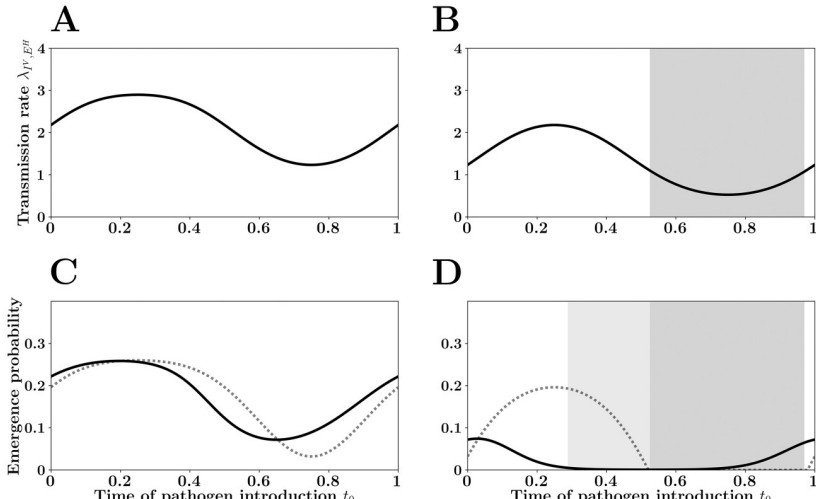

**Fig 5. Probability of Zika emergence across space and time.** The top figures (A and B) show the seasonal variations in $\lambda_{IV,\ E^H}$, the transmission rate from humans to the vectors because of the fluctuations the density of vectors in two habitats (this illustrates the effect of *space* on Zika emergence): a minor variation in mean temperature, 29˚C (A and C) versus 27˚C (B and D), has a massive impact on transmission and, consequently, on pathogen emergence. In C and D we illustrate the effect of the *time* of introduction $t_0$ on Zika emergence. The dotted black line refers to the naive expectation for the probability of pathogen emergence at time $t_0$ if all the rates were constant and frozen at their $t_0$ values (see (7)). The gray shading in B and D refers to the low transmission season where the product of the transmission rates is lower than the product of death rates (see S1 Text). The exact probability of emergence $p_e(t_0\ T, T)$ is indicated as a solid black line. Higher seasonality (B and D) increases the discrepancy between the naive expectation and the exact value of the probability of pathogen emergence. This discrepancy is due to the *winter is coming* effect (light gray shading in D). Parameter values are given in table S1 A (model I) of section 3 of S1 Text.

transition rates of the pathogen life cycle. When the period of the fluctuation is large (i.e., $T > 1/\mu$) the probability of emergence varies with the timing of pathogen introduction. This probability drops when the pathogen is introduced at a point in time where conditions are unfavorable (low transmission and/or high recovery rates). More surprisingly, we show that the probability of pathogen emergence can also be very low in times where conditions are favourable if they are followed by a particularly hostile environment. This *winter is coming* effect results from the existence of adverse conditions that introduce demographic traps (where the net reproduction rate is negative) and pathogen emergence is only possible if the pathogen introduction occurs sufficiently far ahead of those traps. This effect is also expected to act on the size of the epidemics in deterministic models. Epidemics initiated at the end of the high transmission season are expected to be smaller because they do not have time to expand before reaching the low transmission season [26]. There is good evidence of this effect in measles [27].

Note that our approach neglects the density dependence that typically occurs after some time with major epidemics. Our probability of pathogen emergence thus provides an upper approximation of the probability emergence. Indeed, with density dependence the size of the pathogen population may be too small to survive even very shallow demographic traps. In section 5 of S1 Text we show how such density dependence can magnify the *winter is coming* effect.

Understanding this effect allows us to identify the optimal deployment of control strategies minimizing the average probability of pathogen emergence in seasonal environments. We identified optimal control strategies in different epidemiological scenarios under the assumption that the introduction time is homogeneous (Figures 3, 5, and A, D in S1 Text). This

assumption can be readily modified to take into account temporal variations in the probability of introduction events, which yields different recommendations for the timing of control (see subsection 1.3 and Figure C in S1 Text).

This work can be extended to explore optimal timing of other control strategies. For instance [19] study the optimal timing of pulse vaccination in seasonal environment and show for a range of epidemiological scenarios that a pulse vaccination applied periodically 3 months before the peak transmission rate minimizes $R_0$. Yet, as pointed out above, the strategy minimizing $R_0$ may not always coincide with the strategy minimizing $\langle p_e(t_0) \rangle$ (see Fig 4). Indeed, an examination of figure H in S1 Text shows that the probability of emergence is minimized if pulse vaccination occurs a bit sooner than the time at which $R_0$ is minimized (3.71 instead of 3 months before the peak transmission).

So far we focused on control strategies that lower pathogen transmission. Our approach can also be used to optimize control measures that do not act on the transmission rate but on the duration of the infection. For instance, what is the optimal timing of a synchronized effort to use antibiotics to minimize bacterial pathogens emergence? We found that the timing of these treatment days have no impact on $R_0$ but pathogen emergence is minimized when treatment occurs 1.3 months before the peak of the transmission season. This strategy creates deeper traps and results in a stronger *winter is coming* effect. Interestingly, [28] explored the optimal timing of mass antibiotic treatment to eliminate the ocular chlamydia that cause blinding trachoma. Numerical simulations showed that the speed of eradication is maximized (the time to extinction is minimized) when treatment is applied 3 months before the low transmission season. A similar result was obtained by [29] showing that it is best to treat against malaria in the low transmission season. The apparent discrepancy between these recommendations is driven by the use of different objective functions (pathogen emergence, speed of eradication or cumulative number of cases).

The above examples show that our analysis has very practical implications on the understanding and the control of emerging infectious diseases in seasonal environments. This theoretical framework could be used to produce maps with a very relevant measure of epidemic risk: the probability of pathogen emergence across space and time (Fig 5). Currently available risk maps are often based on integrated indices of suitability of pathogens or vectors [30, 31, 32, 33]. These quantities may be biologically relevant but the link between these quantities and the probability of pathogen emergence is not very clear. We contend that using risk maps based on $p_e(t_0)$ would be unambiguous and more informative. Our model could thus contribute to development of "outbreak science" [34] and help public health services to forecast the location and the timing of future epidemics. More generally, the same approach could also be used to improve the prevention against invasions by nonindigenous species [35].

Experimental test of theoretical predictions on pathogen emergence are very scarce because the stochastic nature of the prediction requires massive replicate numbers. Some microbial systems, however, offer many opportunities to study pathogen emergence in controlled and massively replicated laboratory experiments [36]. It would be interesting to use these microbial systems to study the impact of periodic oscillations of the environment to mimic the influence of seasonality. Another way to explore this question experimentally would be to use data on experimental inoculation of hosts. Indeed, the experimental inoculation of a few bacteria in a vertebrate host (which could be viewed as "population" of susceptible cells) is equivalent to the introduction of a few pathogens in a host population. The outcome of these inoculations are stochastic and the probability of a successful infection (host death) is equivalent to a probability of emergence. Interestingly, some daily periodicity to bacterial infections has been found in mice [37, 38]. Mice inoculated early in the morning (4am) have a higher probability of survival than mice inoculated at any other time. This pattern is likely to result from a circadian control

of the vertebrate immune system [39] which are likely to impact the birth and death rates of bacteria. Given that the generation time of a bacteria is smaller than a day, it is not surprising to see a probability of emergence depending on the inoculation time (see Eq 6). In other words, our work may also be used to shed some light on the stochastic within-host dynamics of pathogen infections. One could envision that simple changes in therapeutic practices that take into account the time of day may affect clinical care and could limit the risk of nosocomial infections. Our work provides a theoretical toolbox that can integrate detailed description of the periodic nature of pathogen life cycles at different spatial and temporal scales (within and between hosts, over the period of one day or one year) to time optimal control strategies.

## Methods

### Pathogen emergence with seasonality

The life cycle of a directly transmited pathogen is governed by its birth and death rates ($\lambda$ and $\mu$, respectively). In the absence of seasonality these birth and death rates are constant ($\lambda > 0$, $\mu > 0$), the basic reproduction number is $R_0 = \frac{\lambda}{\mu}$ and the probability of extinction, starting initially with one individual, is $q = \inf\left(1, \frac{1}{R_0}\right)$ (Fig 1). This result was first derived by [40].

In a seasonal environment the birth and death rates are assumed to be functions of time, noted $\lambda(t)$ and $\mu(t)$, respectively, the basic reproduction number is harder to compute but the extinction probability is well known (see e.g. [12] or [13, Chapter 7]). This yields (Eq 2) for $p_e(t_0)$, the probability of pathogen emergence when a single infected host is introduced in the host population at time $t_0$.

Let us now consider rates with period $T > 0$, denoted by $\lambda_T$ and $\mu_T$. Accordingly, we denote $\varphi_T(t) := \int_0^t (\lambda_T(s) - \mu_T(s))\, ds$ and $p_e(t_0, T)$ the corresponding emergence probability. The basic reproduction number has been derived in [3, 14] as the spectral radius of the next generation operator, and is the ratio of time averaged birth and death rates (see Eq (3)). Since $\varphi_T(t) \sim_{t \to +\infty} (\bar{\lambda} - \bar{\mu})t$, we find that $p_e(t_0, T) = 0$ if $R_0 \leq 1$.

If $R_0 > 1$, we can rearrange formula (2) and express $p_e(t_0, T)$ as Eq (4) which varies with the ratio of average birth and death rates, but with a weight that takes into account the average growth rate of the pathogen population. Indeed, first observe that since $\varphi_T'(t) = \lambda_T(t) - \mu_T(t)$ we have

$$\int_{t_0}^t \lambda_T(s)e^{-\varphi_T(s)}\, ds - \int_{t_0}^t \mu_T(s)e^{-\varphi_T(s)}\, ds = \left[-e^{-\varphi_T(s)}\right]_{t_0}^t = e^{-\varphi_T(t_0)} - e^{-\varphi_T(t)}. \tag{8}$$

Since $\varphi_T(t) \to +\infty$ this implies

$$\int_{t_0}^\infty \lambda_T(s)e^{-\varphi_T(s)}\, ds = e^{-\varphi_T(t_0)} + \int_{t_0}^\infty \mu_T(s)e^{-\varphi_T(s)}\, ds. \tag{9}$$

We now use periodicity to obtain, first that for integer $k$,

$$\varphi_T(t + kT) = \varphi_T(t) + k\int_0^T \varphi_T(s)\, ds = \varphi_T(t) + kT(\bar{\lambda} - \bar{\mu}), \tag{10}$$

and thus

$$\int_{t_0}^{\infty} \lambda_T(s) e^{-(\varphi_T(s)-\varphi_T(t_0))} \, ds = \sum_{k=0}^{+\infty} \int_{t_0+kT}^{t_0+(k+1)T} \lambda_T(s) e^{-(\varphi_T(s)-\varphi_T(t_0))} \, ds \tag{11}$$

$$= \sum_{k=0}^{+\infty} \int_{t_0}^{t_0+T} \lambda_T(s) e^{-(\varphi_T(s)-\varphi_T(t_0)-kT(\bar{\lambda}-\bar{\mu}))} \, ds \tag{12}$$

$$= \frac{1}{1 - e^{-T(\bar{\lambda}-\bar{\mu})}} \int_{t_0}^{t_0+T} \lambda_T(s) e^{-(\varphi_T(s)-\varphi_T(t_0))} \, ds \,. \tag{13}$$

Similarly,

$$\int_{t_0}^{\infty} \mu_T(s) e^{-(\varphi_T(s)-\varphi_T(t_0))} \, ds = \frac{1}{1 - e^{-T(\bar{\lambda}-\bar{\mu})}} \int_{t_0}^{t_0+T} \mu_T(s) e^{-(\varphi_T(s)-\varphi_T(t_0))} \, ds \,. \tag{14}$$

Hence,

$$p_e(t_0, T) = \frac{1}{\int_{t_0}^{\infty} \lambda_T(s) e^{-(\varphi_T(s)-\varphi_T(t_0))} \, ds} = \frac{1 - e^{-T(\bar{\lambda}-\bar{\mu})}}{\int_{t_0}^{t_0+T} \lambda_T(s) e^{-(\varphi_T(s)-\varphi_T(t_0))} \, ds} \tag{15}$$

and

$$p_e(t_0, T) = 1 - \frac{\int_{t_0}^{\infty} \mu_T(s) e^{-(\varphi_T(s)-\varphi_T(t_0))} \, ds}{\int_{t_0}^{\infty} \lambda_T(s) e^{-(\varphi_T(s)-\varphi_T(t_0))} \, ds} \tag{16}$$

$$= 1 - \frac{\int_0^T \mu_T(s+t_0) e^{-\varphi_T(s+t_0)} \, ds}{\int_0^T \lambda_T(s+t_0) e^{-\varphi_T(s+t_0)} \, ds} \,. \tag{17}$$

## Asymptotic results for small and large periods

Under the assumption that $R_0 = \frac{\bar{\lambda}}{\bar{\mu}} > 1$ we know that $p_e(t_0, T) > 0$ for all $t_0$. In the following we rescale time so that the $T$ periodic functions $\lambda_T$, $\mu_T$ become 1 periodic functions defined by

$$\lambda(t) := \lambda_T(tT), \quad \mu(t) := \mu_T(tT) \,. \tag{18}$$

And similarly,

$$\varphi(t) = \int_0^t (\lambda(s) - \mu(s)) \, ds = \frac{1}{T} \varphi_T(tT) \,. \tag{19}$$

Hence, the introduction time $t_0$ refers to introduction time between 0 and 1 and by a change of variables we obtain

$$p_e(t_0 T, T) = \frac{1 - e^{-T(\bar{\lambda}-\bar{\mu})}}{T \int_0^1 \lambda(s+t_0) e^{-T(\varphi(s+t_0)-\varphi(t_0))} \, ds} \,, \quad (t_0 \in [0, 1]) \,. \tag{20}$$

In the following we derive simpler expressions for $p_e(t_0 T, T)$ in the limit cases where $T$ is very small or very large.

## Asymptotics for small periods: When $T \to 0$

We see from Eq (20) that when $R_0 = \frac{\bar{\lambda}}{\bar{\mu}} > 1$ that we have

$$\lim_{T \to 0} p_e(t_0 T, T) = \frac{\bar{\lambda} - \bar{\mu}}{\bar{\lambda}} = 1 - \frac{1}{R_0} \, . \tag{21}$$

In other words when $T \to 0$, we can replace the varying rates by their means. Indeed we have on one hand, as $T \to 0$,

$$1 - e^{-T(\bar{\lambda} - \bar{\mu})} \sim T(\bar{\lambda} - \bar{\mu}) \, . \tag{22}$$

On the other hand, since $\lambda$ has period 1,

$$\int_0^1 \lambda(s + t_0) e^{-T(\varphi(s+t_0) - \varphi(t_0))} \, ds \sim \int_0^1 \lambda(s + t_0) ds = \int_0^1 \lambda(s) \, ds = \bar{\lambda} \, . \tag{23}$$

## Asymptotics for large periods: When $T \to +\infty$

We observe on various examples that for large $T$, $p_e(t_0 T, T)$ can sometimes be very small on subintervals of $[0, T]$.

We are going to give a mathematical formulation to this observation. Define

$$\pi_e(t_0) := \begin{cases} 1 - \frac{\mu(t_0)}{\lambda(t_0)} & \text{if } \lambda(t_0) > \mu(t_0) \, ; \\ 0 & \text{if } \lambda(t_0) \leq \mu(t_0) \, . \end{cases} \tag{24}$$

to be the *guess* we make for large periods by substituting in the formula giving the emergence probability for constant rate $\lambda(t_0)$ and $\mu(t_0)$ to $\lambda$ and $\mu$. It is natural to define the *winter period*, $W$, as

$$W = \{t_0 : \lambda(t_0) \leq \mu(t_0)\} \, . \tag{25}$$

However, the period where the emergence probability is vanishingly small is larger than $W$. We call this interval (or set of intervals) *WIC* (for *Winter Is Coming*) and we have (see Proposition 6.1 of the section 6 of the S1 Text):

$$\lim_{T \to +\infty} p_e(t_0 T, T) = \begin{cases} 1 - \frac{\mu(t_0)}{\lambda(t_0)} & \text{if } t_0 \notin WIC \, , \\ 0 & \text{if } t_0 \in WIC \, , \end{cases} \tag{26}$$

with

$$WIC = \{t_0 \in [0, 1) : \lambda(t_0) \leq \mu(t_0) \text{ or } \exists s > t_0, \varphi(s) \leq \varphi(t_0)\} \, . \tag{27}$$

In other words a time $t_0$ is in the *WIC* interval if it is already in $W$ (*winter period*) or if there is a demographic trap in the future. A demographic trap occurs if there is a time $s > t_0$ for which the expected size of the population $X(s)$ at time $s$ is smaller than the original size at the introduction time $X(t_0)$:

$$\mathbf{E}[X(s) \mid X(t_0) = x_0] = x_0 e^{\varphi(s) - \varphi(t_0)} \leq x_0 \, . \tag{28}$$

## Supporting information

**S1 Text. This document contains complementary material that supports the results that we discuss in the main body of the paper.** We present: (1) a calculation of the probability of emergence of directly transmitted pathogen for different scenarios of seasonality, (2) a generalisation of our results when the pathogen life cycle goes through multiple stages before completing its life cycle, (3) an exploration of the *winter is coming* effect on the seasonal dynamics of Zika virus, (4) an analysis of a scenario that involves pulse interventions (vaccination or treatment), (5) an exploration of the effect of density dependence on the *winter is coming* effect, (6) additional computations and proofs.
(PDF)

## Acknowledgments

We thank Mike Boots and Sébastien Lion for comments on an earlier draft. Philippe Carmona thanks the Centre Henri Lebesgue ANR-11-LABX-0020-01 for creating an attractive mathematical environment.

## Author Contributions

**Conceptualization:** Philippe Carmona, Sylvain Gandon.

**Formal analysis:** Philippe Carmona.

**Investigation:** Philippe Carmona.

**Software:** Philippe Carmona, Sylvain Gandon.

**Supervision:** Sylvain Gandon.

**Visualization:** Philippe Carmona, Sylvain Gandon.

**Writing – original draft:** Philippe Carmona, Sylvain Gandon.

**Writing – review & editing:** Philippe Carmona, Sylvain Gandon.

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
