## [Decision Letter · Decision Letter 0]

3 Feb 2020

Dear Dr Gandon,

Thank you very much for submitting your manuscript 'Winter is coming: pathogen emergence in seasonal environments' for review by PLOS Computational Biology.  We apologize for the long time of the process and for the un-satisfactory point we have reached. Your manuscript was evaluated by only one referee so far, and the delay was due to a long process of trying to secure at least a second referee. After having invited 13 others, we would like to move forward on the basis of this referee and the evaluation by one of us.

The reviewer indicated the importance of the problem, but raised concerns on the presentation of the arguments and results, and the way the reader would have to construct an understanding of the main idea. We agree with this concern and think that the way the argument is presented may have been itself one reason why we have failed to secure referees (rather than a lack of interest in the subject). 

While your manuscript cannot be accepted in its present form, we are willing to consider a revised version in which the issues raised by the reviewer have been adequately addressed. We believe a clearer version will have a better chance of additional reviews, which we would like to have to make a decision. We cannot promise publication at that time, but we believe this is a more effective course of action than seeking those reviews now. 

Please return the revised version within the next 60 days. If you anticipate any delay in its return, we ask that you let us know the expected resubmission date by email at ploscompbiol@plos.org. Revised manuscripts received beyond 60 days may require evaluation and peer review similar to that applied to newly submitted manuscripts.

Sincerely,

Mercedes Pascual

Associate Editor

PLOS Computational Biology

Virginia Pitzer

Deputy Editor

PLOS Computational Biology

[LINK]

Reviewer's Responses to Questions

**Comments to the Authors:**

Reviewer #1: see attached

**Have all data underlying the figures and results presented in the manuscript been provided?**

Reviewer #1: Yes

PLOS authors have the option to publish the peer review history of their article (what does this mean?). If published, this will include your full peer review and any attached files.

Reviewer #1: No

---

## [Decision Letter · Decision Letter 1]

6 May 2020

Dear Dr. Gandon,

Thank you very much for submitting your manuscript "Winter is coming: pathogen emergence in seasonal environments" for consideration at PLOS Computational Biology.

As with all papers reviewed by the journal, your manuscript was reviewed by members of the editorial board and by two referees.  The first referee had commented on the previous version of the manuscript and had requested clarifications and changes to make the work more accessible. She/he is now satisfied with the changes and provides minor suggestions. The second referee agrees that the work is a valuable and interesting contribution.  But  his/her major concerns are again with the way the work is presented, in particular the lack of  sufficient context motivating the work and placing its relevance into a more tangible and historical context. Specific suggestions are made for the consideration of a number of studies that would help doing this.  We happen to agree that the manuscript can be written in a way to reach a broader audience and goes from the technical contribution to deeper context and discussion.

In light of this review (below this email) and our own assessment, we would like to invite the resubmission of a revised version that takes into account the reviewer's comments in a way that makes the results of the work more compelling. We cannot make any decision about publication until we have seen the revised manuscript.

Thank you again for your submission and your patience with the delays in the current circumstances.

We hope you will consider this additional request a constructive attempt to make the manuscript as accessible and appealing as possible to our readers.

Please don't hesitate to contact us if you have any questions or comments.

Sincerely,

Mercedes Pascual

Associate Editor

PLOS Computational Biology

Virginia Pitzer

Deputy Editor

PLOS Computational Biology

Reviewer's Responses to Questions

**Comments to the Authors:**

Reviewer #1: This is a very carefully worked out and mature work that covers a lot of ground from periodically forced natural systems, to their optimal control. The authors have a detailed knowledge of stochastic systems and branching processes, and this has led to the fruition of a deep analysis from many fronts. I learnt a lot from reading this manuscript.

1) The authors have made significant new contributions to the study of forced seasonal systems and highlighted some new (sometimes subtle) important dynamical behavious.

2) I have checked all the maths in the SI that is relevant to the equations and models discussed in the main text.

Minor comments:

*Authors should mention that Eqns 5&6 are derived in the relevant Appendix.

*Page 5 Eqn.5 Perhaps give the value of R0, and pE in the main text, for Fig.2E and 2F so the reader can see this works.

*Ref missing in Section 2.3 of SI

*Fig.2 The notation p(toT, T) is used in the SI, but I don't think it is given in the main text. (I am referring here to the rescaling of t0)

Reviewer #2: The topic and idea developed in this work is very interesting, and I find it particularly valuable that the authors emphasize the importance of working with probabilities rather than with deterministic quantities such as R0. Although R0 is useful for many purposes, it presents limitations, especially for low-transmission situations, among others.

Although the development of expressions and the study of probabilities for the occurrence of major outbreaks are not new, there is a lack of concrete applications and discussion of different transmission scenarios. This theoretical study investigates how the interaction of seasonality, duration and arrival time of the infection, affects the probability of a major outbreak. In particular, the authors contribute to the formalization of an effect that is somehow intuitive which they call “the winter is coming effect”.

Unfortunately, however, the authors do not better exploit the intuitive and biological aspects of this effect in a way that would be accessible to a broader audience (that would be very interested in this topic). The way the paper is written is quite technical. It would benefit from a deeper connection to particular infectious diseases or actual transmission results, mainly in the introduction and discussion sections. For example, the authors could compare/connect their results with Otero et al. 2010, who studied the probabilities of major dengue outbreaks based on the first infection arrival time to the city of Buenos Aires. Another example that could be discussed in this paper is the study of flu arrival times for different cities in the US and UK by Truscott & Ferguson 2012, as well as the results obtained by Dalziel et al. 2018 also for flu. In addition, I was surprised that the authors did not cite and connect their probabilities’ expressions with those in the papers by Bartlett 1964 and by Lloyd et. al 2007. Both papers discuss the probability of pathogen emergence, in direct and vector transmitted diseases.

In conclusion, I like the idea and I think it is a compelling topic that a broad spectrum of readers of this journal and beyond would be interested in. The writing is too technical and needs deeper motivation and discussion related to empirical and intuitive aspects, including a clearer connection to results in other studies of transmissible diseases.

**Have all data underlying the figures and results presented in the manuscript been provided?**

Reviewer #1: Yes

Reviewer #2: Yes

PLOS authors have the option to publish the peer review history of their article (what does this mean?). If published, this will include your full peer review and any attached files.

Reviewer #1: No

Reviewer #2: No
---

## [Editor Report · Decision Letter 2]

15 May 2020

Dear Dr. Gandon,

We are pleased to inform you that your manuscript 'Winter is coming: pathogen emergence in seasonal environments' has been provisionally accepted for publication in PLOS Computational Biology.

Best regards,

Mercedes Pascual

Associate Editor

PLOS Computational Biology

Virginia Pitzer

Deputy Editor

PLOS Computational Biology

---

## [Editor Report · Acceptance letter]

26 Jun 2020

PCOMPBIOL-D-19-01772R2

Winter is coming: pathogen emergence in seasonal environments

Dear Dr Gandon,

I am pleased to inform you that your manuscript has been formally accepted for publication in PLOS Computational Biology. Your manuscript is now with our production department and you will be notified of the publication date in due course.

With kind regards,

Laura Mallard
